# Integrating design-of-experiments (DOE) optimization and risk assessment towards a safe and simplified electroporation protocol for *Toxoplasma gondii*

**Pratik Narain Srivastava**[1,2], **Nicholas Perewernycky**[1,2,3], **Lucca Filippo**[4,5], **Lianne M. Lefsrud**[4], **Mark Ungrin**[1,2,6]*

**1** Faculty of Veterinary Medicine, University of Calgary, Calgary, Alberta, Canada, **2** Alberta Children's Hospital Research Institute, University of Calgary, Calgary, Alberta, Canada, **3** Biomedical Engineering Graduate Program, University of Calgary, Calgary, Alberta, Canada, **4** David and Joan Lynch School of Engineering Safety and Risk Management, University of Alberta, Edmonton, Canada, **5** Department of Mechanical Engineering, University of Alberta, Edmonton, Canada **6** Department of Biomedical Engineering, Schulich School of Engineering, University of Calgary, Calgary, Canada,

* mdungrin@ucalgary.ca

## Abstract

Genetic manipulation of *Toxoplasma gondii* presents unique challenges due to its obligatory intracellular nature and relatively rapid growth. Electroporation is the main technique used to introduce genetic modifications into *T. gondii*. However, the existing protocols require an electroporation buffer comprised of eight components and involving multiple steps of preparation. Optimizing electroporation protocols, including a readily available buffer is crucial for achieving efficient transfection while simplifying the overall process. In this study, we present a modified Opti-MEM I based electroporation buffer that matches cytomix in performance with significantly reduced variability. We also develop a novel scoring method (etScore) to reproducibly quantify electroporation performance, combining transgene gene expression with cell viability. We also couple the experimental work with a corresponding systematic risk assessment and argue for routine use of such tools in similar contexts. We anticipate this protocol will make genetic modification of *T. gondii* more accessible to the international community, accelerating drug and vaccine research.

## Author summary

In this study, we report a novel electroporation buffer that significantly simplifies *Toxoplasma* electroporation by using a convenient, off-the-shelf medium, while maintaining transfection efficiency and reducing experimental variability. Our approach advances the accessibility and scalability of electroporation for *T. gondii* genetic manipulation, thus making drug and vaccine research accessible to a wider scientific community. In addition to simplifying the buffer requirements,

**Data availability statement:** All data used to create graphs and perform statistical analysis/ modelling are available in the supplementary information as a compressed zip file.

**Funding:** This work was supported by an Alberta Innovates (https://albertainnovates. ca/) Postdoctoral Enhancement Fellowship (to PS), an Alberta Children's Hospital Research Institute (https://research4kids.ucalgary. ca/) Graduate Scholarship (to NP), Natural Sciences and Engineering Research Council of Canada (https://nserc-crsng.canada.ca/en) Discovery Grants RGPIN 2018-05170 (to LL) and RGPIN-2022-03553 (to MU); and Social Sciences and Humanities Research Council of Canada (https://sshrc-crsh.canada.ca/en.aspx) New Frontiers in Research Fund – Exploration Grant NFRFE-2020-00575 (to MU). The funders had no role in study design, data collection and analysis, decision to publish, or preparation of the manuscript.

**Competing interests:** The authors have declared that no competing interests exist.

we also provide a comprehensive risk assessment of our protocol using standardized methods and suggest steps to ensure the safety of laboratory personnel while working with a pathogen like *T. gondii*. This streamlined protocol and detailed safety analysis reduces active time, lowers operational barriers, and makes it suitable for high-throughput applications.

## Introduction

Transfection of DNA and RNA is a vital research technique, increasingly relevant in the clinical context as well [1–3]. *Toxoplasma* Historically, DNA transfection has utilized biological methods, primarily viral vectors, though not available for apicomplexans like *Toxoplasma gondii* [4], and chemical methods, offering alternatives to viral approaches [5]. These chemical methods include lipid–nucleic acid complexes, cationic lipids containing cyclen and ammonium moieties, and polyethylenimine-based gene vectors [6]. While generally considered safer and more controllable than biological methods [4], these chemical approaches often require significant incubation times with target cells, a limitation due to *T. gondii* tachyzoites' susceptibility to viability loss in serum-free media at room temperature [7]. Consequently, physical methods of gene delivery remain state-of-the-art for non-mammalian eukaryotic cells such as *T. gondii*. These include optical transfection, magnetofection, biolistic transfection, micro-injection, and electroporation [6]. Electroporation, favored for its accessibility and cost-effectiveness, contrasts with micro-injection's high expense and labor intensity [6].

*Toxoplasma* genetic manipulation evolved significantly from 1970-80, establishing in vitro tachyzoite culture, chemical mutagenesis, selection, and cloning via limiting dilution [8]. Simultaneously, protocols for parasite sexual reproduction and genetic crossing in cats were developed [9]. "Forward genetics" combined these methods to investigate nucleotide biosynthesis, virulence factors, population structure, and *T. gondii* evolution [10]. "Reverse genetics" approaches, emerging during 1980–90, pioneered *Toxoplasma* transgenesis techniques, including insertional mutagenesis and gene replacement at loci like dihydrofolate reductase thymidylate synthase (TgDHFR/TS) and uracil phosphoribosyltransferase (TgUPRT), enabling precise gene editing and elucidating parasite pathways [11,12]. More recently, CRISPR/Cas9 technology has become standardized for *Toxoplasma*, facilitating insertional tagging, marker rescue, and further insights into gene function, virulence, and host-parasite interactions [13]. Throughout this advancement, electroporation has served as a foundational method for DNA delivery. As a major rate-limiting step in research into *Toxoplasma* genetics, it is not only critical to achieve high electroporation efficiency but also ease of use and protocol simplicity.

Previous research for optimizing electroporation conditions for *T. gondii* focused on a particularly prevalent buffer named cytomix. Its complexity arises from eight components, including pH-balanced salt mixtures mimicking mammalian cytosol and additives like ATP, EGTA/EDTA, and glutathione resulting in variability and challenges

in standardization [14]. While cytomix demonstrated improved cell survival compared to standard buffers like PBS or cell culture media [14], early studies reported comparatively low parasite viability [12]. Other protocols using Lonza Nucleofector systems have also been developed. Electroporation buffers for these systems are not only proprietary but also tied to specific instruments with undisclosed parameters and protocols [15–19]. Therefore, we did not include these in our buffer comparison and optimization study. Interestingly, studies of other transfection methods, such as lipofection, have shown that common cell culture media (e.g., DMEM) can negatively impact efficiency when compared to Opti-MEM I (OM, Gibco, 11058021) [20,21]. OM has been shown to be superior to PBS and DMEM for transfection in various cell types [20,21]. This paper investigates the potential of Opti-MEM I (mOM) as a novel base medium for *Toxoplasma* electroporation, addressing the inherent complexity and variability associated with cytomix.

Beyond the research itself, the growing capacity for biological system manipulation necessitates proactive consideration of broader implications, particularly Laboratory Acquired Infections (LAIs) involving pathogens such as *Toxoplasma* [22]. While systematic reviews offer general risk information and mitigation controls (e.g., WHO risk group classification and BSL assignment), they often lack detailed assessment of institutional-level controls [23]. We advocate for the routine integration of risk identification and mitigation strategies into process development. To this end, we apply modern engineering risk management tools, presenting our findings for adoption while highlighting the availability of these methods and the value of their consistent application.

## Results

### A robust electroporation performance score (etScore) derived from RT-qPCR data

Accurate analysis of RT-qPCR depends on the correct estimation of fluorescence and baseline calculation of PCR efficiency from sample data. Traditionally, analysis software packages supplied with real time PCR systems rely on a constant or user-defined baseline assignment which may bias the results [24]. On the other hand, PCR efficiencies calculated from sample serial dilutions are also subject to experimental variability by the virtue of inadvertent sample transfer and other user errors. Although some of these errors can be circumvented by using pipetting robots, the process is inherently susceptible to noise. To overcome these issues and develop a robust transfection efficiency scoring function, linear regression-based estimation of fluorescence baseline along with inline PCR efficiency calculations were used in all data analysis [24]. LinRegPCR, an assumption-free RT-qPCR data analysis tool was used for uniform baseline estimation, PCR efficiency determination and subsequent analysis of starting quantities of CAT and TgGAPDH from electroporated and non-electroporated samples [25]. Electroporation efficiency was calculated as fold change (log2FC) in CAT expression normalized to TgGAPDH expression, and parasite viability post electroporation was calculated as percent viability using equations (1) and (2), respectively.

$$log2FC = \log_2\left(\frac{\frac{CAT_T}{TgGAPDH_T}}{\frac{CAT_C}{TgGAPDH_C}}\right)$$

(1)

$$pVia = \frac{TgGAPDH_T}{TgGAPDH_C} \times 100$$

(2)

Where,

$CAT_T$ = CAT expression in transfected samples.

$TgGAPDH_T$ = TgGAPDH expression in transfected samples.

$CAT_C$ = CAT expression in control samples.

$TgGAPDH_C$ = TgGAPDH expression in control samples.

log2FC = log2 Fold change in gene expression.

pVia = Percentage parasite viability.

Electroporation efficiency and parasite viability values were standardized using equations (3) and (4). Min-max standardization was used instead of z-score to prevent scaling differences introduced by differing standard deviations of individual datasets and applying a common boundary condition, making the datasets more suitable for modelling and machine learning applications [26].A constant value of 1 was added post-standardization to avoid division by 0 in subsequent calculations.

$$Std_{log2FC} = \left( \frac{log2FC - min(log2FC)}{max(log2FC) - min(log2FC)} \right) + 1 \tag{3}$$

$$Std_{pVia} = \left( \frac{pVia - min(pVia)}{max(pVia) - min(pVia)} \right) + 1 \tag{4}$$

$$etScore = \frac{2}{\left( \frac{1}{Std_{log2FC}} + \frac{1}{Std_{pVia}} \right)} \tag{5}$$

Where,

$Std_{log2FC}$ = Standardized log2FC.

$Std_{pVia}$ = Standardized percentage viability.

etScore = Electroporation performance score.

## Small scale *T. gondii* electroporation can be efficiently detected by RT-qPCR

Counting fluorescent *T. gondii*, either manually under a microscope or automatically using flow-cytometry, has been the standard method to determine electroporation efficiency. Transgene expression can reliably be detected approximately after 8 hours of electroporation and accurate efficiency calculations depend on the marker being robust and simple to detect. Recently, a Taqman-quantitative PCR approach to quantify site-specific mutagenesis was developed for *T. gondii*, which provides the basis for an automated, reliable and reproducible estimation of transfection efficiency [27]. In this study, RT-qPCR was utilized to assess relative electroporation efficiency and parasite viability by quantifying CAT and TgGAPDH expression.

Traditionally, tachyzoites and plasmid DNA required for stable transfection of *T. gondii* range between $10^6$ to $10^7$ and 50–100 µg, respectively [28]. To test the quantity of plasmid DNA needed to efficiently detect electroporation by RT-qPCR, 1, 5 and 10 µg plasmid DNA was used to electroporate $1\times10^5$ tachyzoites (Table 1). Electroporations were performed in complete cytomix as described above. CAT mRNA expression ranging from 90 to 3000-fold (represented as log2FC) was detected with no significant difference in average expression between the three tested plasmid DNA amounts (Fig 1A).

Table 1. Tachyzoite numbers and plasmid DNA amounts tested for RT-qPCR based detection of CAT expression after electroporation.

| Determination of tachyzoite number | | Determination of plasmid DNA | |
|---|---|---|---|
| Tachyzoites (x10⁵) | Plasmid DNA (µg) | Tachyzoites (x10⁵) | Plasmid DNA (µg) |
| 1 | 5 | 1 | 1 |
| 2 | 5 | 1 | 5 |
| 5 | 5 | 1 | 10 |

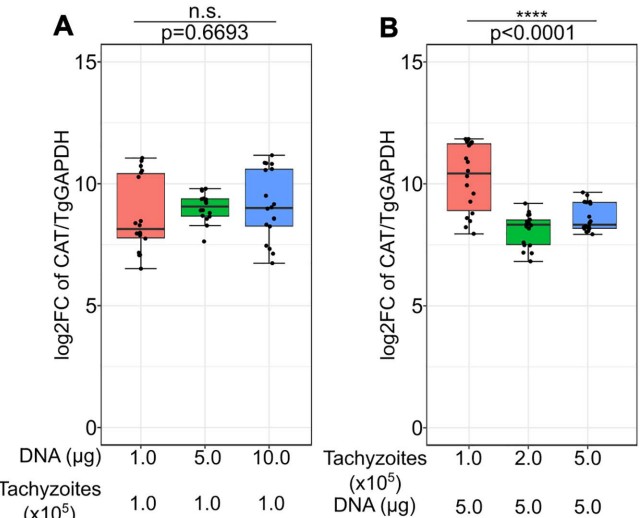

**Fig 1. Levels of CAT expression in(A) 1x10⁵ tachyzoites electroporated with 1, 5 and 10 µg plasmid DNA.Most homogeneous expression was observed using 5 µg plasmid DNA as examined by Levene's test (p = 0.006583).** Otherwise, no significant difference was found between mean values. And (B) 1, 2 and 5x105 tachyzoites electroporated with 5 µg plasmid DNA. Significantly higher CAT expression was observed in the lowest number of tachyzoites with a steep decline with increasing numbers. Data was collected from 3 independent experiments with 6 technical replicates each. Box-whisker plots depict the median and quartile range of individual datasets, and p-values were derived from Welch's one-way ANOVA (Tables A and B in S1 Data).

It was also observed that using 5 µg DNA resulted in the most consistent performance (coefficient of variation = 6.8%) as compared to 1 µg (coefficient of variation = 13.5%) and 10 µg (coefficient of variation = 11.9%), which was further confirmed by comparing the spread between the 3 groups using Levene's test [29]. Subsequently, to test the effect of increasing tachyzoite numbers on transfection, 1, 2 and 5x10⁵ tachyzoites were electroporated with 5 µg DNA. A significant decrease in CAT mRNA expression was observed with increasing tachyzoites. While 1x10⁵ tachyzoites/5 µg DNA had similar expression values as stated above, about 19% reduction in CAT expression was observed in both 5x10⁵ and 10x10⁵ tachyzoites (Fig 1B). Therefore, a combination of 1x10⁵ tachyzoites and 5 µg plasmid DNA was used for further experiments and electroporation performance scoring.

## ATP and EDTA have a significant effect on the electroporation performance of mOM

The original composition of cytomix contains ATP (2 mM), EDTA (2 mM) and GSH (5 mM) as additives. These are not a part of MEM composition, on which OM, and hence the proposed mOM buffer, are based. Therefore, the effect of these additives on electroporation performance of mOM was quantified by using a discretized 2-level screening experimental design (Table 2).

When compared with unsupplemented OM, addition of 2 mM ATP resulted in 32.8% ± 4.4% (p-value < 0.001) increase in efficiency; 2 mM EDTA resulted in a 22.5% ± 2.7% (p-value < 0.001) gain, while 5 mM GSH resulted in a mere 3.6% ± 6.0% (p-value = 0.8165) higher electroporation performance. Even combined with either ATP or EDTA, addition of GSH did not result in a meaningful performance improvement over ATP or EDTA (22.9% ± 1.7%, p-value = 0.0210 and 19.0% ± 1.9%, p-value = 0.8967 respectively). Conversely, combination of ATP and EDTA without GSH resulted in a significant performance increment of 41.8% ± 5.9% over stock OM (p-value < 0.001). All p-values were derived from Games-Howell post-hoc pairwise comparison of means resulting from Welch's one way ANOVA (Fig 2A). Data from all screening experiments

**Table 2. 2-level screening design to determine the effects of ATP, EDTA and GSH concentrations on mOM performance.**

| Factor | Level 1 (mM) | Level 2 (mM) |
|---|---|---|
| ATP | 0 | 2 |
| EDTA | 0 | 2 |
| GSH | 0 | 5 |

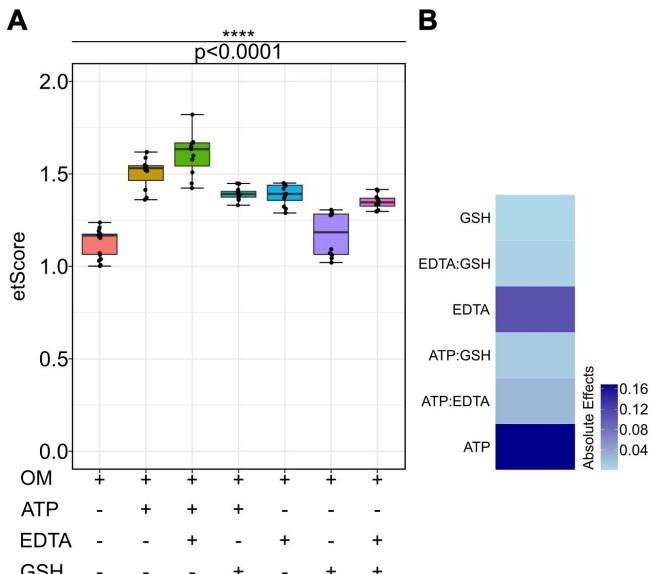

**Fig 2. Effect of ATP, EDTA and GSH on the performance of mOM as an electroporation buffer. (A)** Significant increase in etScore was observed after adding ATP and/or EDTA but not GSH. Data was collected from 3 independent experiments with 6 technical replicates each. Box-whisker plots depict the median and quartile range of individual datasets, and p-values were derived from Welch's one-way ANOVA. **(B)** Absolute effects of ATP, EDTA, GSH and their pairwise interactions on etScore also show that GSH doesn't have a major role to play in the capacity of mOM as an electroporation buffer (Table C in S1 Data).

was combined to fit a weighted linear regression model for assessing the absolute effects of these additives and their possible interactions on the electroporation performance of mOM (Equation 6, S2 and S3 Tables).

$$etScore = \beta_0 + \beta_1 ATP + \beta_2 EDTA + \beta_3 GSH + \beta_4(ATP)(EDTA) + \beta_5(ATP)(GSH) + \beta_6(EDTA)(GSH) \tag{6}$$

A heatmap of absolute effects clearly indicated that both ATP and EDTA have a significant but independent effect on etScore, while the effects of their interaction as well as GSH and its interactions were much less pronounced (Fig 2B, S2 Fig).

## Optimization of ATP and EDTA concentrations in mOM

Prolonged exposure to ATP has been found to be cytotoxic at 5 mM in a human cervical cancer cell culture [30] and while EDTA is not generally cytotoxic, it can disrupt membrane integrity, induce single or double stranded breaks in cellular DNA and increase reactive oxygen species even at very low concentrations [31]. Levels of ATP were therefore centered at 7 mM, with the lower extreme being 0 and the upper being 14.1 mM, and EDTA at 2 mM with the extremes being 0.6 and

**PLOS** **Neglected Tropical Diseases**

3.4 mM. A 2-factor orthogonal central composite experimental design (CCD) was used to optimize the selected additive concentrations (Fig 3A and 3B). CCD requires the acquisition of "star" data points lying on the circumference of a circle with the radius, $\alpha = \sqrt{2}$ along with central and factorial data points. Real values of factor levels (ATP and EDTA concentrations) were coded on a 2D plane as (0,0) for the central points, (-1,-1), (+1,+1), (-1,+1) and (+1,-1) for factorial points, and (0,+α), (0,-α), (+α,0) and (-α,0) for star points (Table 3 and Fig 3B). Star points represent univariate extremes in factor levels and depend on the choice of center points. CCD helps avoid the need to test all possible factor combinations by providing a rational method of data analysis using the response surface methodology (RSM). However, using a conventional RSM, including linear, quadratic and interaction terms has been shown to be less effective at modeling complex processes [32]. Therefore, cubic terms were included in the model after iteratively analyzing their performance in reducing prediction errors (Equation 7, S4 and S5 Tables). Additional models using just the quadratic terms and quadratic interaction terms in addition to cubic terms were also tested. Our balanced model ($R^2$ = 0.9847, Adjusted $R^2$ = 0.9798, Lack of fit p-value = 0.2764) performed significantly better than the quadratic only model ($R^2$ = 0.9594, Adjusted $R^2$ = 0.9509, Lack of fit p-value = 6.101x10$^{-05}$) while not over-fitting like a full cubic model with quadratic interaction terms ($R^2$ = 0.9855, Adjusted

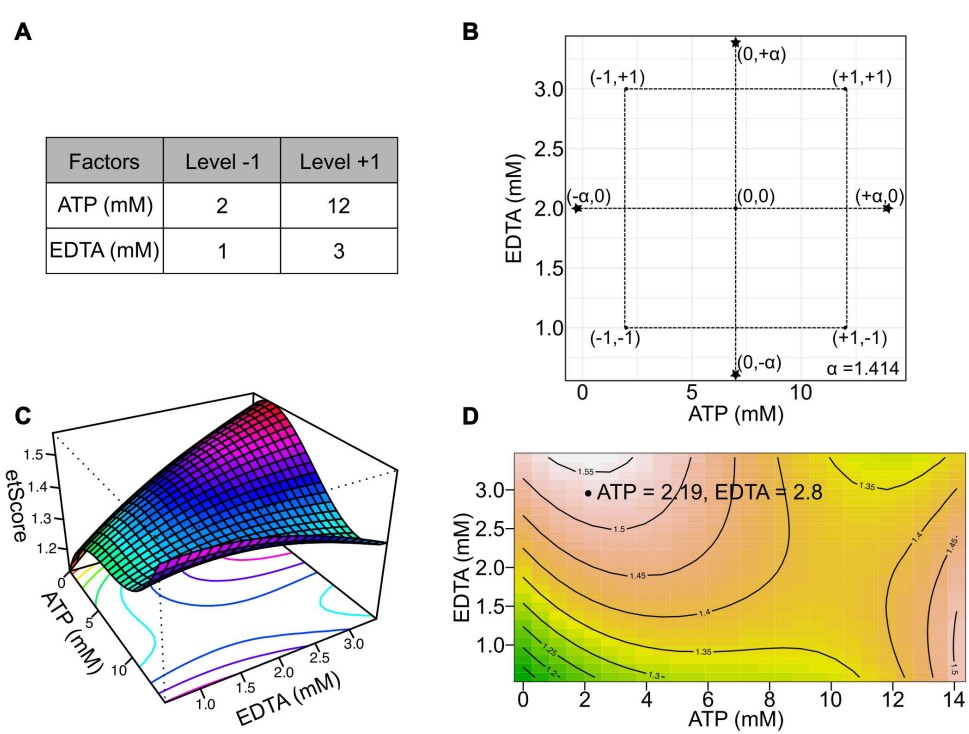

**Fig 3. (A) Levels of ATP and EDTA used for the CCD experiment. (B)** Experimental design depicting axial, star and center points. 6 replicates for the center point and 3 replicates for each star points were utilized. Data was collected from 3 independent experiments. **(C)** 3D response surface plot depicting the change in etScore with ATP and EDTA concentrations. **(D)** Contour plot depicting the working concentrations of ATP and EDTA required to achieve maximum etScore value in these experiments (Table D in S1 Data).

**Table 3. Central composite experimental design to optimize the concentrations of ATP and EDTA for mOM electroporation.**

| Factors | Levels (mM) | | | | |
|---|---|---|---|---|---|
|  | -α | -1 | 0 | +1 | +α |
| ATP | 0 | 2 | 7 | 12 | 14.1 |
| EDTA | 0.6 | 1 | 2 | 3 | 3.4 |

$R^2 = 0.98$, Lack of fit p-value = 0). Additionally, detailed comparison using Akaike Information Criterion (AIC), Bayesian Information Criterion (BIC) and Likelihood Ratio Test (LRT) also showed that our model adequately described the desired effects of ATP and EDTA on mOM's performance (S5-S7 Tables).

$$etScore = \beta_0 + \beta_1 ATP + \beta_2 EDTA + \beta_3 ATP^2 + \beta_4 EDTA^2 + \beta_5(ATP)(EDTA) + \beta_6 ATP^3 + \beta_7 EDTA^3 \qquad (7)$$

The resulting model was validated using standardized residual analysis, 5-fold cross-validation and learning curves (S3 Fig) [33]. A response surface was used to determine optimal additive concentrations required to maximize electroporation performance. Shape of the surface indicated that etScore increased more rapidly with EDTA than ATP. The local maximum of the surface within the experimental range was observed under low ATP and high EDTA concentration (Fig 3C). The same data are also presented as a 2D contour plot annotated with the concentrations of ATP and EDTA highlighting the local maximum of the etScore value (Fig 3D). To simplify reagent preparation along with optimal electroporation performance, we chose the working concentration of ATP at 2 mM (same as cytomix) while EDTA was increased to 3 mM.

### Electroporation performance of mOM is similar to cytomix

Electroporation performance of mOM prepared from two batches of OM (ATP, EDTA and GSH added immediately before use) was compared to cytomix prepared from two different stock salt solutions (Using different sources of ultrapure water. ATP, GSH and CaCl2 added immediately before use) in 4 independent experiments. $10^5$ tachyzoites were electroporated with 5 µg pTub:dYFP-pSAG1:CAT in complete cytomix and mOM. Electroporation efficiency and parasite viability were calculated using RT-qPCR, standardized and combined as etScore, which was observed to be similar for both groups (Fig 4). Levene's test revealed a significant improvement in consistency, with mOM yielding approximately a 2.7-fold reduction in variance (p = 0.0055) as compared to cytomix.

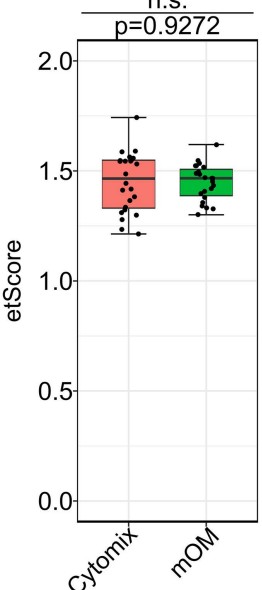

**Fig 4. Electroporation performance (etScore) comparison of mOM with cytomix.** No significant difference was observed between the two buffers, but mOM was found to be more reproducible and consistent between batches of OM as compared to cytomix prepared from different batches of stock solutions (Levene's test, p = 0.0055). Data was collected from 4 independent experiments with 6 technical replicates each. Box-whisker plots depict the median and quartile range of individual datasets, and p-values were derived from Welch's one-way ANOVA (Table E in S1 Data).

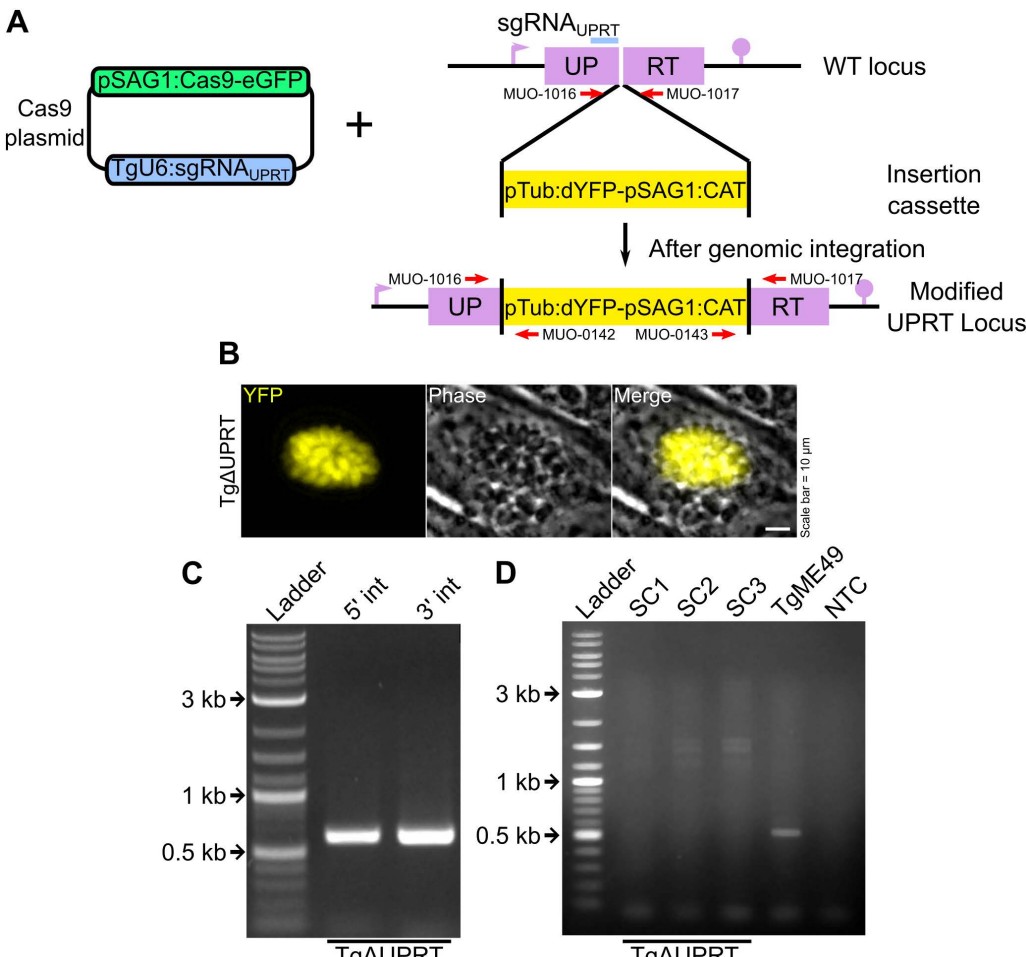

**Fig 5. Using mOM as an electroporation buffer for stable genetic modifications in *T. gondii*. (A)** Schematic representation of gene insertion strategy at the TgUPRT locus using a Cas9 induced double stranded break combined with a repair cassette with a selectable marker (CAT)/reporter (dYFP) system. **(B)** YFP expression in TgΔUPRT indicates successful cassette insertion. **(C)** Diagnostic PCRs using site specific primers confirm targeted cassette integration. **(D)** Additional diagnostic PCRs to confirm the clonality of 3 TgΔUPRT colonies and disruption of the TgUPRT locus.

## mOM is suitable for stable genetic modification of *T. gondii*

TgUPRT is a non-essential genomic locus in *Toxoplasma* which is amenable to stable genetic modification [13]. The gene was knocked out to create a strain (TgΔUPRT) with a stably integrated selection marker/reporter cassette (linearized pTub:dYFP-pSAG1-CAT) by electroporating it along with a plasmid expressing spCas9 using mOM as the electroporation buffer (Fig 5A). Fluorescent parasites were observed 24 h after transfection (Fig 5B). TgΔUPRT parasites were selected by growing them on chloramphenicol until spontaneous egress of fluorescent parasites. Site specific integration PCRs were performed which showed successful integration of the insertion cassette on both 5' and 3' ends without the need of homology regions (Fig 5C). Single clonal lines were isolated by embedding TgΔUPRT infected HFF monolayers in agarose and subsequently excising single colonies. No wild type contamination was observed in the three isolated clonal lines (SC1, SC2 and SC3) as tested by amplification of a small region around the Cas9 target site, with a band appearing only in TgME49 parental, wild type strain (Fig 5D).

## Systematic safety analysis of *T. gondii* culture and genetic modification protocols

As the technical methods in this study facilitate the genetic modifications of a pathogen, their potential for associated workplace and community safety hazards were assessed, even when working with attenuated laboratory strains using bowtie analysis (BTA). BTA is a widely-used risk assessment method that enables the identification and mitigation of systemic failures [34–36]. The main benefit of BTA is its ability to generate a clear and easily understandable overview of multiple hazardous scenarios, their associated prevention and mitigation controls and the outcomes that the hazards are likely to generate if left unaddressed [37].

*T. gondii* is a ubiquitous parasite, possibly infecting up to one-third of the global population, while not causing symptoms in otherwise healthy individuals. Known natural routes of infection are ingestion of tissue cysts with under-cooked, contaminated meat or oocysts from contaminated water [38]. Cysts are environmentally stable and not affected by changes in temperature, pH and exposure to bleach [39].

In the context of a research laboratory, percutaneous inoculation and mucous membrane exposure were identified as possible LAI routes (Fig 6). Tachyzoites, the stage of *T. gondii* life cycle studied in vitro, are not environmentally stable with a survival rate of a few hours outside their host cells in laboratory conditions. Despite that, presence of a high concentration in cultures and varying experimental techniques makes them infective by induced exposure. Thus, "Exposure to *Toxoplasma gondii* leading to LAI" was selected as the top event, which allowed the consideration of more mitigations that could be adopted after an exposure and potentially prevent the development of an infection.

Using a syringe needle to shear host cells and liberate tachyzoites for experiments is a major part of the standard protocols, as discussed in the methods section. Skin injuries, either pre-existing or caused by contact with a needle or broken glassware, may provide a route for tachyzoites to infect laboratory personnel. Percutaneous inoculations become even more likely when laboratory personnel are not provided with inherently safe equipment (e.g., blunt needles and cut-resistant gloves) and lack access to adequate protocols and well-structured, hands-on biosafety training. There is also a rare, but significant chance of creating environmentally stable cysts through mishandling of experimental procedures. For example, an old tachyzoite culture started with a low inoculum may have some cysts if the pH of culture medium is above 8. Exposed mucous membranes such as the eyes, nose or oral cavity may come into contact with tachyzoites through high-pressure manipulation of tachyzoite suspension. Simple laboratory actions such as mixing, pipetting or syringe shearing have the potential to create droplets or splashes that may reach exposed mucous membranes. Although, toxoplasmosis is not considered to be airborne in its natural context, this possibility must be considered. Poor ventilation, damaged HEPA filters of a biological safety cabinet combined with high pressure manipulation of *T. gondii* cultures for various experimental procedures may result in aerosol particles falling in the size range of being suspended in the air for a few hours [40].

Three scenarios were selected that could represent outcomes of an accidental *T. gondii* exposure for which multiple preventive barriers can be applied viz. "No infection," "Infection of personnel with or without fatal consequences," and "Infection of personnel with or without fatal consequences resulting in secondary transmission" (Fig 6). Attenuated strains should be used for research wherever possible, as suggested in the WHO laboratory safety manual [41]. *T. gondii* strains unable to form tissue cysts [42] and oocysts [43] due to induced loss-of-function genetic modifications have been reported. These strains can be used for experiments involving drug testing for acute toxoplasmosis. All personnel should undergo medical examination to test for toxoplasmosis before starting to work with *T. gondii*. Currently, there are no vaccines against toxoplasmosis but prophylactic treatments for toxoplasmosis include spiramycin, pyrimethamine-sulphonamide, trimethoprim-sulfamethoxazole, and dapsone-pyrimethamine [44]. These should be included in institutional health and wellness programs. Periodic medical surveillance of laboratory personnel should also be conducted along with enquiring and updating information about emergency medical contacts [45].

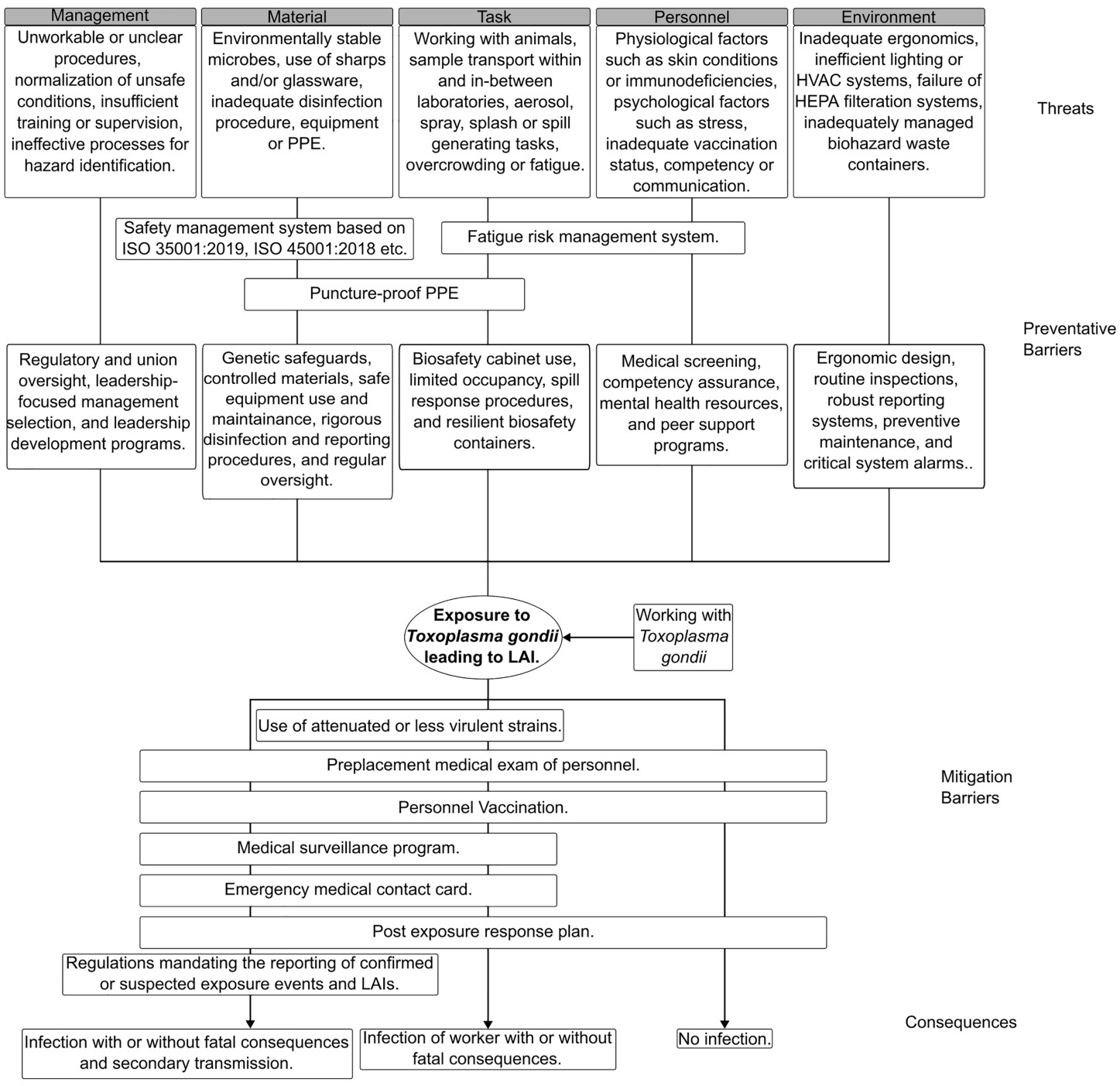

**Fig 6. Bowtie-based systematic analysis of *T. gondii* genetic modification process to help ensure safety against LAIs.**

Finally, proper detection of toxoplasmosis is important to prevent spread and take pre-emptive action. Common detection methods are microscopic and sero-testing while PCR based testing is also becoming more common. Laboratory personnel should be tested regularly as a preventive measure.

## Discussion

In this study, a robust, adaptable and sensitive statistical approach was employed to quantify electroporation performance by combining transgene expression and cell viability. mOM was established as a lower-complexity, less labour-intensive substitute, yielding a 2.7 fold reduction in variance between experiments. Although RT-qPCR was used in this study to quantify transfection efficiency, our scoring metric (etScore) can be generalized to other quantification methods such as fluorescence (S1 Appendix, S4 Fig). Initial experiments established that small scale *Toxoplasma* electroporations could be easily and meaningfully quantified by RT-qPCR with $10^5$ tachyzoites and 1 μg plasmid DNA. RSM enabled us to perform relatively rapid screening experiments to efficiently assess the performance of cytomix and OM supplemented with different levels of ATP, EDTA and GSH. The resulting buffer, mOM had a similar electroporation performance. Furthermore, only ATP and EDTA had statistically significant roles on the performance of mOM. Although the exact composition of OM is not publicly available, we hypothesize that the hypoxanthine present in OM [46] is able to substitute for GSH in preventing post-electroporation oxidative damage. Testing this hypothesis will need further experiments which are beyond the scope of this study, but the results helped us eliminate GSH as an additive and further simplify the buffer. A DOE approach was used to optimize the working concentrations of ATP and EDTA to achieve optimum electroporation performance. A 2-factor CCD was implemented across multiple experiments and analyzed using a weighted RSM. Within boundaries based on previous reports of toxicity [30,31], 2 mM ATP and 3 mM EDTA were found to provide the best electroporation outcome and hence were used for further experiments. To test its efficacy in stable genetic modifications, we employed mOM to knock-out the non-essential TgUPRT locus by introducing a Cas9 mediated break and integrating a YFP reporter/CAT selection marker system. We also provided a systematic safety analysis of *T. gondii* culture and genetic modification procedures. Use of a blunt needle for cell shearing and removal of unnecessary centrifugation and filteration steps were implemented as proper mitigation controls for the same.

## Conclusion

Simplicity and amenability to automation present significant advantages for biomedical laboratory methods – increasing reproducibility, reducing manual labour, and reducing potential for injury to researchers. In this study, widely used protocols for culture and manipulation of *Toxoplasma* were assessed through the lens of our recent study on risk mitigation and laboratory acquired infections [47], and modified to enhance efficiency, simplicity, and risk mitigation. Some steps from the standard *Toxoplasma* electroporation protocol such as tachyzoite filtration before electroporation were also eliminated via the use of larger inoculum in the starter culture and a blunt needle for isolation. This removed most host cell debris effectively, as well as significantly reducing a potential source of needle-stick injury for laboratory workers. A novel scoring system was established for use with response-surface approaches that we hope will be applied to further optimization of related protocols in *T. gondii* as well as other apicomplexans.

This study involves the use of *Toxoplasma gondii* ME49. It is a type II, live virulent strain used as one of the primary models of chronic toxoplasmosis for the purpose of drug and vaccine development. While it is not hyper-virulent like the type I strains (e.g., RH), it is also not avirulent like the type III strains (e.g., CTG). Type II strains are generally classified as intermediate virulent, warranting systematic safety analysis of their manipulation protocols [48].

Our paper represents an initial step towards simplifying *Toxoplasma* genetic modification protocols, making them more accessible to researchers in the international community – a particular concern in laboratories focusing on region-specific parasitic infections that may be of high local priority but lack the commercial potential to attract attention from the pharmaceutical industry [49]. We also hope to encourage adoption of advanced engineering risk management approaches developed over decades in the transport and energy sectors, to better mitigate risks when working with potentially infectious organisms.

## Materials and Methods

### Cells and parasite culture

Human foreskin fibroblast (HFF, CRL-1634) cell line was obtained from the American Type Culture Collection (ATCC) and maintained in DMEM supplemented with 10% FB Essence (Seradigm, 10803–034) or Fetalgro EX (RMBIO, 76644–832). Cells were cultured for not more than 20 passages and used to propagate a type II *Toxoplasma gondii* strain ME49 (a kind gift from Dr. Constance Finney, University of Calgary). Parasites were also cultured for a maximum of 20 passages to maintain genetic uniformity for the transfection experiments. Transgenic parasites were selected after electroporation using 40 µM chloramphenicol when required. All cell line and parasite experiments were approved under Social Sciences and Humanities Research Council of Canada (SSHRC) New Frontiers in Research Fund – Exploration (NFRF-E) Grant NFRFE-2020–00575 by the university of Calgary biosafety committee in accordance with Public Health Agency of Canada (PHAC) risk group II as outlined in Canadian biosafety standards.

### Plasmid and primers

pTub:dYFP-pSAG1:CAT containing two copies of the yellow fluorescent protein (dYFP) in tandem under the *Toxoplasma* alpha tubulin (Tub) promoter as a reporter and chloramphenicol acyltransferase (CAT) under *Toxoplasma* major surface antigen (SAG1) promoter as a selection marker was received as a kind gift from Dr. Boris Striepen, University of Pennsylvania [50]. pSAG1:Cas9-U6:sgUPRT expressing spCas9 under the *Toxoplasma* surface antigen 1 (SAG1) promoter and a small guide RNA targeting the non-essential uracil phosphoribosyl transferase under the U6 promoter was obtained from addgene (https://www.addgene.org/54467/) [13]. RT-qPCR primers were designed for quantifying CAT expression (MUO-0087/MUO-0088) after electroporation and *Toxoplasma gondii* glyceraldehyde 3-phosphate dehydrogenase (TgGAPDH) expression (MUO-0089/MUO-0090) for normalization. Primers used in this, and other experiments are listed in S1 Table.

### Electroporation buffers

All stock solutions required for cytomix were prepared in ultra-pure water (Table 4) as described previously [28]. Incomplete cytomix working solution was formulated by mixing salt components (except $CaCl_2$) at their final concentrations and adjusting its pH to 7.6 with KOH. It was then stored at 4 ∘C for up to three months [51]. To prepare complete cytomix, $CaCl_2$, ATP and GSH (Table 4) were always added immediately prior to use to prevent precipitation or degradation. OM was used as an alternative electroporation buffer and supplemented with ATP, EDTA and GSH as specified below. Several variants of this modified OM (mOM) were assessed (Fig 2, Table 5).

**Table 4. Cytomix (pH 7.6) composition [14].**

| Component | Vendor | Concentration (mM) |
|---|---|---|
| Potassium chloride (KCl) | EMD PX1405–1 | 120 |
| Calcium chloride ($CaCl_2$) | Sigma-Aldrich C7902 | 0.15 |
| Potassium phosphate mono/dibasic ($K_2HPO_4$/$KH_2PO_4$) | Amresco 0705/Amresco 0781 | 10 |
| Magnesium Chloride ($MgCl_2$) | Sigma-Aldrich M8266 | 5 |
| 4-(2-hydroxyethyl)-1-piperazineethanesulfonic acid (HEPES) | ThermoFisher 15630–080 | 25 |
| Ethylenediaminetetraacetic acid (EDTA) | Sigma-Aldrich E7889 | 2 |
| Adenosine triphosphate (ATP) | Enzo ALX-480–021-G005 | 2 |
| Glutathione (GSH) | Sigma-Aldrich G4251 | 5 |

**Table 5. Variants of mOM tested for Toxoplasma electroporation.**

| Variant | ATP (mM) | EDTA (mM) | GSH (mM) |
|---|---|---|---|
| OM | 0 | 0 | 0 |
| mOM-1 | 2 | 0 | 0 |
| mOM-2 | 0 | 2 | 0 |
| mOM-3 | 0 | 0 | 5 |
| mOM-4 | 2 | 2 | 0 |
| mOM-5 | 2 | 0 | 5 |
| mOM-6 | 2 | 2 | 5 |

### *Toxoplasma* electroporation

All electroporation experiments were performed on a Biorad Gene Pulser (Biorad 165–2076) with modified parameters described previously [51]. Briefly, required number of tachyzoites released by shearing infected cells with a 27G blunt needle were resuspended in 150 µl pre-chilled electroporation buffer, mixed with pTub:dYFPpSAG1:CAT and transferred to a 0.4 cm electroporation cuvette (Biorad 1652088) on ice [52]. Electroporation was performed in an open circuit using a single pulse of 1.8 kV and 3 µF without the pulse controller module, resulting in a field strength of 4.5 kV/cm and a time constant of 0.5 ms. Electroporated tachyzoites were allowed to recover at room temperature for 15 minutes before adding onto fresh HFF monolayers grown in 6 well plates.

### Real time quantitative PCR (RT-qPCR)

HFF cells infected with electroporated *T. gondii* were lysed 24 h post-transfection, by directly adding Trizol (Invitrogen 15596026). The lysate was mixed with chloroform (20% by volume), allowed to stand at room temperature for 5 minutes and centrifuged at 13000xg, 4 °C for 10 min. Upper phase was collected and total RNA was precipitated by adding equal volume of cold isopropanol and centrifuging at 13000xg, 4 °C for 20 min. RNA pellet was washed twice with 75% ethanol, dried and resuspended in nuclease-free water. cDNA was prepared by reverse transcribing 2 µg total RNA using High-Capacity cDNA Reverse Transcription Kit (Applied Biosystems, 4368814) as per manufacturer's instructions. RT-qPCR was set-up on an Eppendorf epMotion 5070 pipetting robot using PowerTrack SYBR Green Master Mix (Applied Biosystems, A46109). Amplicons of 181 bp (CAT) and 176 bp (TgGAPDH) were amplified (40 cycles) and detected on StepOne (Applied Biosystems) real time PCR system. Amplification data was exported without any baseline correction for further analysis.

### Assessment of mOM as electroporation buffer via generation of a stable UPRT deficient parasite strain

pTub:dYFP-pSAG1:CAT (linearized using KpnI/NotI) was co-transfected with pSAG1:Cas9-U6:sgUPRT for Cas9 mediated gene replacement at the TgUPRT locus. There were no homology regions on the insertion cassette as Cas9 mediated insertion does not require any [13]. Total 50 µg DNA, with a 4:1 ratio of the linear insertion cassette to Cas9-gRNA plasmid was used in combination with $10^6$ freshly lysed *T. gondii* tachyzoites in mOM for electroporation. The number of tachyzoites and the amount of DNA was obtained by scaling up the values optimized in previous experiments. Electroporated tachyzoites were seeded on fresh HFF monolayer and transgenic parasites (TgYFP) were selected by adding 40 µM chloramphenicol 24 h post electroporation, after observing the appearance of YFP fluorescence. Genomic integration was tested using site-specific integration PCR (5' integration: MUO-1016/MUO-0142; 3' integration: MUO-0143/MUO-1017). Parental TgYFP tachyzoites were passaged onto fresh HFF monolayers by 10-fold serial dilution starting from $10^3$ to 10 tachyzoites per well in a 6 well cell culture plate. Infected monolayers were covered with 0.3% agarose and allowed to grow in 40 µM chloramphenicol. Colony formation was assessed under a fluorescent microscope and single colonies were

isolated by excising pieces of agarose at relevant locations and inoculating onto fresh HFF monolayers. Obtained clonal lines were cultured and cryo-preserved for further experimentation. Clonality of isolated lines were tested by a differential PCR to amplify a small internal region of the UPRT locus around the Cas9 target site using primers specific to the intact locus (MUO-1016/MUO-1017).

## Supporting information

**S1 Fig. A schematic representing the experimental flow in this study.**
(PDF)

**S2 Fig. Main effects on etScore for each factor (ATP, EDTA and GSH) across two levels in a two-level screening design.** Marginal means are calculated by averaging over the levels of other factors in the model.
(PDF)

**S3 Fig. Diagnostic evaluation and cross-validation of the response surface model used to optimize ATP and EDTA concentrations for mOM. (A)** Weighted residuals versus fitted response exhibit a random scatter around zero indicating homoscedasticity. **(B-C)** Frequency distribution and normal Q-Q (weighted) plots suggest a normal distribution of residuals. **(D)** Plotting the residuals against the order of observations reveals no discernible trends or patterns. **(E-F)** Five-fold cross-validation confirms the model's robustness and predictive power. **(G)** Learning curves show that the model's error (RMSE) on both the training and validation datasets converges to similar, low values as more data is incorporated into training.
(PDF)

**S4 Fig. Comparison of mOM and cytomix using fluorescence microscopy using (A) mOM and (B) Cytomix.** Quantification of electroporation outcomes using batch image analysis. (C) Standardized transfection efficiency (percentage of green parasites over red parasites). (D) Standardized viability post electroporation (percentage of red parasites over host cell nuclei, i.e., parasitemia) and (E) etScore calculated as described in the Methods section. p-value derived from Welch's two sample t test.
(DOCX)

**S1 Appendix. Validation of etScore to quantify Toxoplasma electroporation using fluorescence microscopy.**
(DOCX)

**S1 Table. Primers used in this study.**
(DOCX)

**S2 Table. Linear model coefficients table for ATP, EDTA and GSH 2-level screening experiment.**
(DOCX)

**S3 Table. Linear model ANOVA table for ATP, EDTA and GSH 2-level screening experiment.**
(DOCX)

**S4 Table. Response surface model coefficients table for ATP and EDTA concentration optimization.**
(DOCX)

**S5 Table. Response surface model ANOVA table for ATP and EDTA concentration optimization.**
(DOCX)

**S6 Table. Response surface model ANOVA table for ATP and EDTA concentration optimization using a quadratic and a full cubic model for comparison with our balanced model.**
(DOCX)

**S7 Table. Comparison of the 3 models using AIC, BIC and LRT.**
(DOCX)

**S8 Table. Previously reported electroporation protocols for *Toxoplasma* transfection.**
(DOCX)

**S1 Data. (A-E) Data used in this study.**
(ZIP)

## Acknowledgments

We thank Dr. Constance Finney for her valuable inputs and the *Toxoplasma gondii* ME49 strain used in this research. We also thank Dr. Boris Striepen, University of Pennsylvania for providing the plasmid pTub:dYFP-pSAG1:CAT used in this study. Additionally, we also thank Dr. Michael G Lipsett for his valuable discussions on laboratory safety.

## Author contributions

**Conceptualization:** Pratik Narain Srivastava, Nicholas Perewernycky, Lucca Filippo, Lianne M. Lefsrud, Mark Ungrin.

**Data curation:** Pratik Narain Srivastava, Nicholas Perewernycky, Lucca Filippo, Lianne M. Lefsrud, Mark Ungrin.

**Formal analysis:** Pratik Narain Srivastava, Lianne M. Lefsrud, Mark Ungrin.

**Funding acquisition:** Lianne M. Lefsrud, Mark Ungrin.

**Investigation:** Pratik Narain Srivastava, Nicholas Perewernycky, Lucca Filippo, Lianne M. Lefsrud, Mark Ungrin.

**Methodology:** Pratik Narain Srivastava, Nicholas Perewernycky, Lucca Filippo, Lianne M. Lefsrud, Mark Ungrin.

**Project administration:** Lianne M. Lefsrud, Mark Ungrin.

**Resources:** Lianne M. Lefsrud, Mark Ungrin.

**Supervision:** Lianne M. Lefsrud, Mark Ungrin.

**Validation:** Lianne M. Lefsrud, Mark Ungrin.

**Visualization:** Pratik Narain Srivastava, Lianne M. Lefsrud, Mark Ungrin.

**Writing – original draft:** Pratik Narain Srivastava, Nicholas Perewernycky, Lucca Filippo, Lianne M. Lefsrud, Mark Ungrin.

**Writing – review & editing:** Pratik Narain Srivastava, Nicholas Perewernycky, Lucca Filippo, Lianne M. Lefsrud, Mark Ungrin.

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
