## [Decision Letter · Decision Letter 0]

3 Dec 2025

Integrating Design-of-Experiments (DOE) Optimization and Risk Assessment Towards a Safe and Simplified Electroporation Protocol for Toxoplasma gondii

Dear Dr. Ungrin,

Thank you for submitting your manuscript to PLOS Neglected Tropical Diseases. After careful consideration, we feel that it has merit but does not fully meet PLOS Neglected Tropical Diseases's publication criteria as it currently stands. Therefore, we invite you to submit a revised version of the manuscript that addresses the points raised during the review process.

Please submit your revised manuscript within by Feb 01 2026 11:59PM. If you will need more time than this to complete your revisions, please reply to this message or contact the journal office at plosntds@plos.org. Please include the following items when submitting your revised manuscript:

We look forward to receiving your revised manuscript.

Kind regards,

Masoud Foroutan, Ph.D.

Academic Editor

Susan Madison-Antenucci

Section Editor

Shaden Kamhawi

co-Editor-in-Chief

Paul Brindley

co-Editor-in-Chief

**Journal Requirements:**

At this stage, the following Authors/Authors require contributions: Pratik Narain Srivastava, Nicholas Perewernycky, Lucca Filippo, Lianne M. Lefsrud, and Mark Ungrin. Please ensure that the full contributions of each author are acknowledged in the "Add/Edit/Remove Authors" section of our submission form.

- ® on page: 24.

- TM on pages: 2, 5, 23, and 24.

Potential Copyright Issues:

- Figure S1.. Please confirm whether you drew the images / clip-art within the figure panels by hand. If you did not draw the images, please provide (a) a link to the source of the images or icons and their license / terms of use; or (b) written permission from the copyright holder to publish the images or icons under our CC BY 4.0 license. Alternatively, you may replace the images with open source alternatives. See these open source resources you may use to replace images / clip-art:

**Reviewers' Comments:**

Reviewer's Responses to Questions

**Key Review Criteria Required for Acceptance?**

**Methods**

-Are the objectives of the study clearly articulated with a clear testable hypothesis stated?

-Is the study design appropriate to address the stated objectives?

-Is the population clearly described and appropriate for the hypothesis being tested?

-Is the sample size sufficient to ensure adequate power to address the hypothesis being tested?

-Were correct statistical analysis used to support conclusions?

-Are there concerns about ethical or regulatory requirements being met?

Reviewer #1: (No Response)

Reviewer #2: (No Response)

Reviewer #3: (No Response)

**Results**

-Does the analysis presented match the analysis plan?

-Are the results clearly and completely presented?

-Are the figures (Tables, Images) of sufficient quality for clarity?

Reviewer #1: (No Response)

Reviewer #2: (No Response)

Reviewer #3: (No Response)

**Conclusions**

-Are the conclusions supported by the data presented?

-Are the limitations of analysis clearly described?

-Do the authors discuss how these data can be helpful to advance our understanding of the topic under study?

-Is public health relevance addressed?

Reviewer #1: (No Response)

Reviewer #2: (No Response)

Reviewer #3: (No Response)

**Editorial and Data Presentation Modifications?**

Reviewer #1: (No Response)

Reviewer #2: (No Response)

Reviewer #3: (No Response)

**Summary and General Comments**

Reviewer #1:

Dear Authors:

This manuscript proposes a simplified electroporation buffer (mOM) for *T. gondii*, optimized through DOE-based modeling and combined with a bowtie-style risk analysis. The topic is relevant and potentially impactful, especially for resource-limited laboratories. The inclusion of a biosafety framework is commendable. Nevertheless, several sections require clarification, additional data, and improved focus., optimized through DOE-based modeling and combined with a bowtie-style risk analysis. The topic is relevant and potentially impactful, especially for resource-limited laboratories. The inclusion of a biosafety framework is commendable. Nevertheless, several sections require clarification, additional data, and improved focus., optimized through DOE-based modeling and combined with a bowtie-style risk analysis. The topic is relevant and potentially impactful, especially for resource-limited laboratories. The inclusion of a biosafety framework is commendable. Nevertheless, several sections require clarification, additional data, and improved focus., optimized through DOE-based modeling and combined with a bowtie-style risk analysis. The topic is relevant and potentially impactful, especially for resource-limited laboratories. The inclusion of a biosafety framework is commendable. Nevertheless, several sections require clarification, additional data, and improved focus.

**Major Comments**

The introduction clearly presents the need to simplify electroporation, but it lacks a detailed comparison with previously optimized buffers or electroporation media used for *T. gondii* (e.g., cytomix derivatives, commercial alternatives). Please better justify how mOM differs fundamentally rather than marginally from cytomix.(e.g., cytomix derivatives, commercial alternatives). Please better justify how mOM differs fundamentally rather than marginally from cytomix.(e.g., cytomix derivatives, commercial alternatives). Please better justify how mOM differs fundamentally rather than marginally from cytomix.(e.g., cytomix derivatives, commercial alternatives). Please better justify how mOM differs fundamentally rather than marginally from cytomix.

While the design-of-experiments approach is valuable, details on replication, validation, and residual analyses are insufficient. Please clarify:Number of replicates per design point.Whether model assumptions (normality, homoscedasticity) were tested.Justification for inclusion of cubic terms in the response-surface model.Provide R² and adjusted R² values for all models.

The “etScore” is a novel but somewhat arbitrary composite metric. Please validate it against conventional measures (e.g., flow cytometry-based transfection efficiency or fluorescence microscopy quantification). Without such comparison, reproducibility across different detection platforms remains uncertain.The manuscript often states “data from 4 experiments,” but it is unclear whether these represent independent biological replicates or technical repeats. Explicitly define both and adjust the statistical interpretation accordingly.The inclusion of a bowtie-based analysis is excellent, yet it dominates the Results section. Consider moving detailed descriptions to Supplementary Materials and summarizing key findings in a concise figure and table.Specify whether the assessment was performed following institutional biosafety standards (e.g., Canadian Biosafety Standards, WHO LSM 4th ed.), and indicate the biosafety level applied to *T. gondii* handling.handling.handling.handling.Claims about applicability to “other apicomplexans” and “resource-limited settings” are somewhat speculative without experimental support. Please moderate these statements or add evidence of preliminary applicability.Figures 1–4 are described well but lack clarity on statistical dispersion (e.g., SD vs SEM).Include representative microscopy images showing typical electroporation outcomes (fluorescence expression, viability).

**Minor Comments**

Ensure all references follow PLOS NTDs citation format and include DOI links.

Add catalog numbers for all reagents where possible.The acknowledgments should clarify that biosafety evaluation did not involve live virulent strains.Provide details of ethical clearance (if applicable) for *T. gondii* strain use.strain use.strain use.strain use.

Below are detailed linguistic and stylistic comments organized according to the major sections of the manuscript.

<h3>**1. Abstract**</h3> </h3> </h3> </h3>

**Main issues:**

Sentences are too long and contain redundant phrases such as *“Till date,” “This introduces complexity,”* etc.etc.etc.etc.Some statements are descriptive rather than result-focused.The abstract reads more like an introduction; it should emphasize **quantitative findings** and and and and **key outcomes**....

<h3>**2. Introduction**</h3> </h3> </h3> </h3>

**Main issues:**

Excellent historical overview, but it is **too detailed and lengthy**....Paragraph transitions are weak, and the main research gap (complexity and variability of cytomix buffer) is not highlighted early.Frequent tense shifts (present ↔ past) and repetitive linking phrases.

<h3>**3. Materials and Methods**</h3> </h3> </h3> </h3>

**Main issues:**

Overly detailed reagent lists and repetition between tables and text.Verb tense inconsistency (switching between past and imperative).Some methods are written like instructions (e.g., “We suggest against using sharp needles…”), which is stylistically informal.

<h3>**4. Results**</h3> </h3> </h3> </h3>

**Main issues:**

Results are presented clearly but with heavy repetition and over-explanation of equations.The flow of logic is occasionally disrupted by detailed formula definitions that could be moved to *Methods*....Frequent use of “It was observed that…” slows down readability.Some subheadings are too long.

<h3>**5. Discussion**</h3> </h3> </h3> </h3>

**Main issues:**

Strong and comprehensive, but overly dense.Repetitive restatement of results rather than interpretation.Long sentences (30+ words) reduce readability.Some claims about applicability to “other apicomplexans” or “global accessibility” are speculative.

<h3>**6. Conclusion**</h3> </h3> </h3> </h3>

**Main issues:**

The section is strong in content but verbose and somewhat repetitive.Contains multiple speculative statements beyond the presented data.Some long sentences could be split for clarity.

Reviewer #2: The manuscript presents a valuable and well-executed study that addresses a significant practical bottleneck in Toxoplasma gondii research: the complexity and variability of the standard electroporation buffer, cytomix. The development of a simplified, commercially available alternative (modified Opti-MEM, mOM) that performs equivalently with reduced variance is a noteworthy contribution. The integration of a novel scoring metric (etScore) and a formal risk assessment using bowtie analysis further strengthens the manuscript, making it of broad interest to the parasitology and molecular biology communities. The work is generally sound, but several major points require clarification and additional data to fully support the claims and ensure robustness.

Major Comments

- The manuscript effectively demonstrates that mOM works for a CRISPR/Cas9-mediated knockout at the TgUPRT locus. However, to establish mOM as a true replacement for cytomix, its performance should be validated across a wider range of common genetic manipulations. The authors should test mOM for:

- Transient Transfection: While RT-qPCR measures CAT mRNA, does this directly correlate with robust protein expression and fluorescence from a transiently transfected reporter (e.g., YFP) when assessed by microscopy or flow cytometry? A direct comparison of fluorescence levels and the percentage of YFP-positive parasites between cytomix and mOM would be highly convincing.

- Stable Integration via Homologous Recombination: The TgUPRT knockout relies on non-homologous end joining (NHEJ) from a Cas9-induced break. It is crucial to test mOM with a standard stable transfection requiring homologous recombination (e.g., using a different locus or a traditional double-crossover strategy) to ensure it is broadly applicable.

- The etScore is a central innovation. The description of its calculation, particularly the standardization step, needs more justification. Why was a min-max standardization chosen over Z-scoring? The authors state a constant of 1 was added to avoid division by zero, but this constant will disproportionately affect datasets with different ranges. The rationale for this specific mathematical construction should be elaborated.

- The claim that etScore is "easily adapted to different transgene detection methods" (e.g., fluorescence) is not demonstrated. The manuscript would be significantly strengthened by showing that etScore, calculated from flow cytometry data (e.g., %YFP+ cells for efficiency and a viability dye for cell health), yields similar conclusions regarding buffer performance. This is critical for labs that do not use RT-qPCR for routine efficiency checks.

- The bowtie analysis is a commendable addition. However, its presentation feels somewhat disconnected from the specific protocol improvements made. The text should more explicitly link the risk analysis to the specific methodological changes adopted. For instance:

- The switch to a blunt needle for host cell shearing is an excellent, concrete outcome of the risk assessment. This should be highlighted as a direct mitigation for the "Percutaneous Inoculation" threat in the bowtie diagram and discussed prominently in the main text.

- Are there other specific protocol steps (e.g., avoiding aerosol-generating steps identified in the risk assessment) that were modified as a result of this analysis? Explicitly stating these would integrate the two halves of the paper more effectively.

- For the CCD optimization, the authors use a complex model with cubic terms. The justification for this over a standard quadratic model is that it reduced prediction errors, but the evidence for this (e.g., comparison of AIC/BIC or R² values between models) is not provided in the main text or supplementary tables (S4, S5). This justification should be clearly stated and the model diagnostics (Fig S3) should be briefly interpreted in the main text to bolster confidence in the model's validity.

- In Figure 1A, the authors use Levene's test to select 5 µg DNA based on homogeneity of variance, but then proceed with Welch's ANOVA, which does not assume equal variances. The logic here is circular. If variance homogeneity is the primary goal, stating that is fine, but the statistical narrative could be clearer.

Minor Comments

1. Introduction:

- The introduction is comprehensive but could be more focused. Consider tightening the discussion on general transfection methods (biological, chemical) and placing greater emphasis on the specific history and documented drawbacks of cytomix in T. gondii research.

2. Results:

- Figure 1: The Y-axis label in 1A and 1B should be "log2FC of CAT/TgGAPDH" for absolute clarity.

- Section 3.2: When stating that GSH had no significant effect, it would be useful to report the p-value from the statistical model for the GSH factor to quantify the lack of effect.

- Section 3.4: The sentence "mOM yielding approximately a 2.7-fold reduction in variance" is clear, but the subsequent statement of a "63% reduction in variance between experiments" in the Discussion appears to be a different calculation (1 - (1/2.7) ≈ 0.63). Please use one consistent metric for clarity, preferably the fold-reduction.

3. Methods:

- Electroporation Parameters: The parameters (1.8 kV, 3 µF) are stated, but the resulting field strength (kV/cm) for the 0.4 cm cuvette should be calculated and mentioned, as this is a standard metric for comparing electroporation protocols.

- RT-qPCR: It is stated that "Amplicons of approx. 180 bp were amplified". Please confirm the exact amplicon sizes for both CAT and TgGAPDH in the methods or a supplementary table.

- Stable Transfection: For the stable knockout, the total DNA amount is 50 µg for 10^6 tachyzoites. This is a high DNA-to-parasite ratio. A comment on why this ratio was chosen, or a comparison to standard protocols using cytomix with this amount, would be helpful.

4. Figures and Tables:

- Figure 3C/D: The contour plot (3D) is excellent. The 3D surface plot (3C) is less informative and could be moved to the supplement to save space.

- Figure 6 (Bowtie Diagram): This figure is dense and critical. In its current form, the font is very small and will be illegible in a published PDF. The authors must significantly improve the readability of this figure, potentially by simplifying the text or re-designing it across multiple panels.

- Table 5: The variants of mOM are clearly listed. It would be helpful to also include the final, optimized mOM buffer (OM + 2mM ATP + 3mM EDTA) in this table for a quick reference.

Typos and Editorial Suggestions

• Abstract: "to across the international community" -> "to the international community"

• Introduction, Page 9, Line 85: "Crispr/Cas9" -> "CRISPR/Cas9"

• Results, Page 11, Line 113: "Table 1" is referenced, but the composition is described in the text on Page 10. Ensure table reference is correct.

• Results, Page 14, Line 190: "1, 2 and 5x10^5 tachyzoites" in the text, but the table header says "Tachyzoites (x10^5)" with values 1, 2, 5. This is inconsistent with the figure legend for 1B which says "1, 5 and 10x10^5". Please check and correct this discrepancy.

• Discussion: "scale-out experiments" – consider using "screening experiments" or "optimization experiments" for clarity.

Reviewer #3:

This manuscript presents a well-executed and thoughtfully designed study that integrates Design-of-Experiments optimization with a detailed safety and risk-assessment framework to develop a simplified electroporation buffer (mOM) for Toxoplasma gondii. The work addresses a clear methodological bottleneck in Toxoplasma genetics: the complexity, batch-variability, and labor-intensive nature of the cytomix buffer. The authors’ use of RT-qPCR–based quantification, their introduction of the etScore metric, and the systematic DOE-based optimization of ATP and EDTA concentrations together represent a strong and innovative approach. The manuscript is generally well-written, methodologically rigorous, and the results are convincingly presented. The inclusion of a bowtie-based risk analysis is a notable strength that elevates the translational and practical value of the work.

Before acceptance, I suggest the authors clarify several methodological points to further strengthen transparency and reproducibility. For example, while the rationale for eliminating GSH is reasonable, the manuscript would benefit from a brief discussion comparing OM’s hypoxanthine levels with typical intracellular antioxidant conditions to support this interpretation. The electroporation parameters are described clearly; however, a concise comparison with the most widely used protocols (e.g., those in the 2020 Methods in Molecular Biology volume) would help contextualize performance improvements. In the safety section, the bowtie diagrams are valuable but could be strengthened by specifying which controls the authors currently implement in their own laboratory versus which are recommended aspirationally for other settings. The conclusion would also benefit from a short statement on anticipated limitation, such as whether mOM performance may vary across Toxoplasma strains or CRISPR constructs.

Overall, this is a strong and impactful manuscript that introduces a practical advancement likely to benefit many laboratories working with T. gondii and other apicomplexans. With minor clarifications and expansion of a few interpretive points, the manuscript will be suitable for publication.

PLOS authors have the option to publish the peer review history of their article (what does this mean?). If published, this will include your full peer review and any attached files.). If published, this will include your full peer review and any attached files.). If published, this will include your full peer review and any attached files.). If published, this will include your full peer review and any attached files.

...

Reviewer #1: No

Reviewer #2: No

Reviewer #3: No

**Figure resubmission:**
---

## [Editor Report · Decision Letter 1]

25 Mar 2026

Dear Dr. Ungrin,

We are pleased to inform you that your manuscript 'Integrating Design-of-Experiments (DOE) Optimization and Risk Assessment Towards a Safe and Simplified Electroporation Protocol for Toxoplasma gondii' has been provisionally accepted for publication in PLOS Neglected Tropical Diseases.

Best regards,

Masoud Foroutan

Academic Editor

Susan Madison-Antenucci

Section Editor

Shaden Kamhawi

co-Editor-in-Chief

Paul Brindley

co-Editor-in-Chief

---

## [Editor Report · Acceptance letter]

Dear Dr. Ungrin,

We are delighted to inform you that your manuscript, "Integrating Design-of-Experiments (DOE) Optimization and Risk Assessment Towards a Safe and Simplified Electroporation Protocol for Toxoplasma gondii," has been formally accepted for publication in PLOS Neglected Tropical Diseases.

Best regards,

Shaden Kamhawi

co-Editor-in-Chief

Paul Brindley

co-Editor-in-Chief
